# Treatment Response in Individual Organs Affected by Chronic Graft-Versus-Host Disease

**DOI:** 10.3390/cells14040238

**Published:** 2025-02-07

**Authors:** Takanobu Morishita, Paul J. Martin, Yoshihiro Inamoto

**Affiliations:** 1Department of Blood and Marrow Transplantation & Cellular Therapy, Fujita Health University School of Medicine, Toyoake 470-1192, Japan; 2Division of Clinical Research, Fred Hutchinson Cancer Center, Seattle, WA 98109, USA

**Keywords:** keyword allogeneic hematopoietic cell transplantation, chronic graft-versus-host disease, organ response, extracorporeal photopheresis, ibrutinib, ruxolitinib, belumosudil, axatilimab

## Abstract

Chronic graft-versus-host disease (GVHD) occurs in 30–70% of patients after allogeneic hematopoietic cell transplantation (HCT) and increases the risks of morbidity and mortality. Systemic corticosteroids are the standard initial treatment, but one-third of patients require subsequent treatment with other systemic agents. Treatment decisions are often based on physicians’ experience. The expected treatment response rates in specific organs affected by chronic GVHD may inform such decisions. In this review, we identify 20 studies reporting treatment response rates in individual organs according to objective criteria, summarize the results, discuss the caveats in data interpretation, identify the unmet needs, and suggest future directions in the field. For cutaneous sclerosis, we observed large discrepancies in organ response rates according to the current NIH criteria and patient-reported improvement, highlighting the need for better measurement tools. High response rates for lung involvement with certain novel drugs deserve further investigation.

## 1. Introduction

Allogeneic hematopoietic cell transplantation (HCT) is a curative option for malignant and non-malignant hematological disorders. Chronic graft-versus-host disease (GVHD) occurs in 30–70% of patients and is one of the major complications affecting morbidity and mortality after HCT [1,2]. Systemic corticosteroids are the standard initial treatment for chronic GVHD, but at least one-third of patients require subsequent-line treatment due to progressive GVHD, an inadequate treatment response, or an intolerance to treatment [3,4,5]. Extracorporeal photopheresis (ECP) has been used in many countries for corticosteroid-refractory or dependent chronic GVHD, and the Japanese regulatory authority has approved its use for this indication [6,7]. Within the past 7 years, the US FDA has approved four drugs (ibrutinib, ruxolitinib, belumosudil, and axatilimab) for second- or subsequent-line treatment for corticosteroid-refractory or dependent chronic GVHD, and regulatory authorities in other countries have also approved some of these drugs for these indications [8,9,10,11].

ECP is an autologous cellular therapeutic strategy for corticosteroid-refractory chronic GVHD. Although the mechanisms of ECP effects have not been fully elucidated, ECP modulates cytokine production, increases the numbers of regulatory T cells (Treg) [12], ameliorates cutaneous and extracutaneous GVHD symptoms, and helps to reduce steroid doses [6,13]. Ibrutinib inhibits Bruton tyrosine kinase (BTK) and the downstream signaling pathway, which activates B cells and regulates their survival. Further, ibrutinib inhibits interleukin-2–inducible T-cell kinase (ITK), which activates T-cell immune responses [8]. In preclinical models, ibrutinib delayed the progression of chronic GVHD and improved disease manifestations [14,15]. Ruxolitinib is a Janus kinase (JAK)1-JAK2 inhibitor. Preclinical studies have shown that JAK1-JAK2 signaling promotes the production of inflammatory cytokines and activates adaptive and innate immune responses by T cells, dendritic cells, and neutrophils in chronic GVHD [9,16]. Belumosudil inhibits Rho-associated coiled-coil-containing protein kinase 2 (ROCK2). Preclinical studies have demonstrated that inhibiting ROCK2 shifts the balance of Th17/Treg cells toward Treg cells, thereby restoring immune homeostasis. Additionally, it suppresses fibroblasts, which helps reduce the inflammatory and fibrotic manifestations of chronic GVHD [17,18,19]. Axatilimab is a humanized IgG4 monoclonal antibody that binds to the CSF-1R extracellular domain and inhibits CSF1R-dependent monocyte activation and profibrotic M2-like macrophage differentiation [20,21]. This differentiation sustains inflammation and tissue injury, leading to maladaptive tissue repair and fibrosis [22,23,24].

In real-world practice, the choices for the treatment of chronic GVHD are based on physicians’ experience, ease of use, risk of toxicity, availability of clinical trials, and possible interference with the anti-tumor effects of donor cells [25,26,27]. The expected treatment response rate in a particular organ may inform such decisions. In this review, we summarize the treatment response rates in individual organs from selected published reports, discuss the caveats in data interpretation, and identify the unmet needs in the field.

## 2. Methods

### 2.1. Selection of Reports

We searched the Pub Med database to identify studies reporting treatment response rates in individual organs affected by chronic GVHD. Reports that met both of the following criteria were included: (1) Both prospective and retrospective studies of second- or subsequent-line treatment for chronic GVHD: Prospective studies were limited to those published between 2005 and 2024. The quality scores of retrospective studies before 2014 were suboptimal [28]. Therefore, retrospective studies were limited to those published between 2014 and 2024. (2) Studies reporting organ response rates according to clearly defined response criteria: The initial search was performed by using terms “graft-versus-host-disease”, and each individual drug among “Ruxolitinib”, “Ibrutinib”, “Extracorporeal photopheresis”, “Belumosudil”, or “Axatilimab.” Review articles, case reports, meta-analyses, non-English reports, and articles without available full text were excluded. The selection of reports was performed according to the PRISMA guidelines [29].

### 2.2. Reporting Quality Assessment of Selected Reports

We used 10 previously proposed quality indicators to evaluate the selected studies [28,30]. Indicators included eligibility criteria, minimization of selection bias, consistent treatment regimen, objective response criteria, overall response criteria, time of assessment, concomitant treatment, historical benchmark, statistical hypothesis, and survival curve. Two individuals (T.M. and Y.I.) independently reviewed reports and evaluated whether each criterion was met or not. Differences in assessments were reconciled by joint review to arrive at a consensus. A total quality score was calculated based on the number of indicators that were met in each report.

### 2.3. Data Extraction of Selected Reports

We extracted data on the best response rate at 24 weeks after administration of each drug. If information was not reported, we extracted fixed-point response rate at 24 weeks. When these endpoints were not reported, we extracted the most relevant data available on response rate and added explanations in footnotes. Pooled response rates in individual sites and drugs were calculated by adding the number of responders and denominators from individual reports separately for best response rate at 24 weeks and for fixed-point response rates at 24 weeks. Studies that did not report either of these endpoints and two reports of combination therapy with multiple drugs were excluded in the pooled response rates.

## 3. Results

### 3.1. Selection of Reports

The initial search as of 24 October 2024 identified 271 reports on ECP, 49 on ibrutinib, 205 on ruxolitinib, 21 on belumosudil, and 4 on axatilimab (Figure 1). The title screening was performed for these 550 records. Records of no HCT data, no chronic GVHD data, wrong population, wrong topic, wrong publication type, and retrospective study before 2014 were excluded. Subsequently, 113 full-text articles were reviewed for eligibility. Studies without a diagnosis of chronic GVHD, those without information on the organ response rate, and those reporting on the initial systemic therapy were excluded. Finally, a total of 20 studies were selected for the review. Ruxolitinib (n = 8) was the most frequently reported drug, followed by ibrutinib (n = 3), ECP (n = 3), belumosudil (n = 4), and axatilimab (n = 2). Two of the eight ruxolitinib studies reported the results of combination therapy that added ECP or belumosudil.

### 3.2. Quality Assessment of Reports

Quality scores for individual reports are shown in Figure 2. The rates of meeting the indicator were suboptimal for two indicators (the historical benchmark and statistical hypothesis), while the rates were good for other indicators. The mean total quality score for all 20 reports was 8.0 (standard deviation 2.1), which was much higher than the 2.0 in the reports of mycophenolate mofetil as a second- or subsequent-line treatment for chronic GVHD that were reported between 1990 and 2011 [28], indicating that the reporting quality has improved considerably since then. Prospective studies have higher total quality scores compared with retrospective studies (mean score 8.9 versus 6.0, respectively, *p* = 0.002).

Table 1 summarizes the results of the studies that we selected for a detailed review. The sections below summarize the studies grouped according to the individual agent that was evaluated.

### 3.3. ECP

Three prospective studies were included in this review. The schedule of ECP therapy differed among the studies: Study #1: three times during week 1, and twice weekly during weeks 2 through 12 [6]; Study #2: twice weekly during weeks 1 through 4, once every other week during weeks 5 through 12, and once monthly during weeks 13 through 18 [32]; and Study #3: three times during week 1, twice weekly during weeks 2 through 12, and twice weekly during weeks 16, 20, and 24 [7]. Notable adverse events associated with ECP include vasovagal reactions, thrombocytopenia, venous access site infection, and fatigue (Table 2).

The median response rate was the highest in the skin (50%), followed by the joints and fascia (44%), GI tract (42%), mouth (41%), liver (36%), eyes (29%), and lungs (16%). The results should be carefully interpreted, because the response criteria differed between these studies, and information on sclerotic skin manifestations was available in only one of them.

In one randomized phase 2 trial, the skin response was evaluated according to changes in a scoring algorithm that assessed 10 topographic areas according to five disease manifestations [6]. In this study, the complete or partial response rate in the ECP arm was 40%, compared with 10% in the control arm. Another study [32] selected patients who had cutaneous sclerotic lesions. The skin response rate according to physicians’ assessment was 83%. The high rate of response in this open-label study should be interpreted with caution because assessment was not based on the changes in objective scores but on the physicians’ perception of change at the follow-up visits. The most recent report described a prospective phase 2 study of ECP [7] and showed a low response rate of 27% in the skin based on a composite endpoint of changes in the NIH skin score and steroid doses at 24 weeks after therapy.

### 3.4. Ibrutinib

Three prospective studies were included in this review. One study focused on pediatric cases [35]. Ibrutinib 420 mg once daily was administered for adult and pediatric patients aged ≥ 12 years in these three studies. Notable adverse events associated with ibrutinib include muscle cramp, diarrhea, nausea, bleeding/bruising, infection, fatigue, and arrhythmia (Table 2). The median best response rate after 24 weeks of therapy according to the NIH criteria among the three studies was highest in the GI tract (81%), followed by the liver (71%), esophagus (63%), mouth (61%), joints and fascia (59%), skin (56%), eyes (29%), and lungs (23%). The high response rate in the liver was supported by the results of a recent retrospective study of 270 real-world patients [46].

Considering the involved immune cells and mechanisms of action, ibrutinib mainly targets B lymphocytes. Autoimmune-like hepatitis is a subtype of liver chronic GVHD, pathologically characterized by the dominant infiltration of plasma cells and B lymphocytes in the portal and periportal sites of the liver [47]. While this observation does not apply to all cases of liver chronic GVHD, it might explain why ibrutinib showed a high response rate for liver chronic GVHD [46].

### 3.5. Ruxolitinib

Two prospective studies and five retrospective studies were included in this review. In the two prospective studies, patients received ruxolitinib at a dose of 10 mg twice daily. Notable adverse events associated with ruxolitinib include anemia, thrombocytopenia, neutropenia, diarrhea, and infection (Table 2). All organs showed ≥50% response rates in at least one study according to the NIH response criteria. The results require careful interpretation as three studies reported response rates at 24 weeks of therapy, while others reported the “best” response rates within a specific time frame. Not surprisingly, response rates were generally higher when measuring the best response compared to the response at 24 weeks. Both prospective studies reported the response rates in individual organs at 24 weeks of therapy. Furthermore, two studies reported the results of ruxolitinib combined with ECP or belumosudil [41,42], although the response rates in these studies did not appear to differ appreciably from the studies of ruxolitinib monotherapy.

Among the three studies reporting response rates at 24 weeks, the median response rate was high in the gastrointestinal tract (54%), esophagus (53%), mouth (50%), and joints and fascia (45%), but lower in the liver (30%), skin (28%), eye (26%), and lung (10%). Among the three retrospective studies reporting the best response rates, the median response rate was high in the gastrointestinal tract (94%), mouth (82%), liver (81%), skin (79%), joints and fascia (78%), and eye (73%), but lower in the lung (50%).

Particularly low response rates in the skin were observed in two studies. One study reported the results of a prospective phase 2 study for patients with refractory sclerotic chronic GVHD [36]. Although the proportion of patients with skin sclerosis was not reported, the skin response rate at 24 weeks of ruxolitinib treatment according to the 2014 NIH criteria was 19%, while the skin response rate according to the Lee symptom subscale was 38%, indicating a need for more consistent, reliable, and sensitive response measures for cutaneous sclerosis [48].

### 3.6. Belumosudil

Four prospective studies were included in this review [10,43,44,45]. In the first study, patients received belumosudil 200 mg once daily or twice daily [10]. In the second study, patients received belumosudil 200 mg once daily, 200 mg twice daily, or 400 mg once daily [43]. In the third and fourth studies, patients received belumosudil 200 mg once daily [44,45]. Notable adverse events associated with belumosudil include fatigue, headache, and infection (Table 2). The median best response rate was high in the joints and fascia (70%), GI tract (69%), mouth (66%), esophagus (58%), and liver (52%), but lower in the skin (40%), eyes (34%), and lungs (17%). Despite highly morbid patients, the high response rate in the joints and fascia was notable. Fibrosis accounts for a substantial part of chronic-GVHD-associated morbidity, particularly in sites such as the joints and fascia [49]. Notably high response rates of 71% [10], 73% [43], 56% [44], and 80% [45] were reported for the joints and fascia in studies of belumosudil. Moreover, patients reported decreased symptom burdens as assessed by the clinically significant improvement in the Lee Symptom Scale (LSS) in these studies with a result of 61% [10], 50% [43], 33% [44], and 57% [45].

Although the best response rates in the lungs ranged from 0% to 26% [10,43,44,45], a combined analysis of two prospective studies with an extended follow-up duration showed that the response rates in the lung were 50% in patients with a baseline score of 1, 15% in those with a baseline score of 2, and 0% in those with a baseline score of 3 [50]. These results suggested that early intervention may be necessary for patients with lung GVHD.

### 3.7. Axatilimab

Two prospective studies were included in this review [11,22]. In the first study, patients received axatilimab at a dose of 0.15 mg/kg, 0.5 mg/kg or 1 mg/kg every 2 weeks, or 3 mg/kg every 2 or 4 weeks [22]. In the second study, patients received axatilimab at a dose of 0.3 mg/kg, or 1 mg/kg or 3 mg/kg every 2 weeks [11]. Notable adverse events associated with axatilimab include fatigue, infection, lipase elevation, transaminase elevation (by inhibiting clearance mediated by Kupffer cells), creatine phosphokinase elevation, and orbital edema (Table 2). The median number of prior treatment lines was four in both studies, and the best overall response rates were 67% and 74%, respectively. The median organ response rates were very high in the GI tract (93%), intermediate in the joints and fascia (69%), esophagus (68%), and mouth (56%), and low in the lung (39%), liver (30%), eye (29%), and skin (21%). The low response rate in the skin is remarkable, but the results require careful interpretation as more than 90% of patients with skin GVHD had sclerotic lesions in one of the studies [11]. Furthermore, when the response was measured by the body surface area of non-movable sclerosis, 44% of patients experienced improvement, and, when the response was measured by patient-reported outcomes, 66% of patients showed an improvement in skin tightening, and 73% of patients showed a reduction in symptoms related to skin tightening. Once more, these results indicate an unmet need for better measures of skin sclerosis. Of note, despite extensive prior treatment, the lung response rate was notably favorable in this study.

### 3.8. Pooled Response Rates for Individual Organs According to Drugs

We extracted data from twelve studies reporting the best response rates at 24 weeks and five studies reporting the fixed-point response rates at week 24 and calculated pooled response rates (Figure 3). The best response rates for the skin and eyes with ruxolitinib were substantially higher than those with other drugs and appeared to stand out to a lesser extent for the mouth, liver, and lung. These observations, however, require careful interpretation for several reasons. First, the four studies that evaluated the best responses to ruxolitinib all had retrospective designs, whereas the studies of all other agents had prospective designs. Second, one prospective study of belumosudil [10] and the two prospective studies of axatilimab [11,22] included significant proportions of patients with skin sclerosis (27% [10], 89% [22], and >90% [11], respectively). In contrast, the proportions of patients with skin sclerosis were not reported in the studies of ruxolitinib, and one prospective study of ibrutinib [8] recruited only patients with an erythematous rash without sclerotic manifestations. The 24-week period of observation is long enough to detect changes in erythema but may not be long enough to detect changes in cutaneous sclerosis. Third, the proportions of severe cases were lower in the ruxolitinib studies (66% [38], 35% [39], and 38% [40]) than in the studies of other drugs: ibrutinib (47% [34] and 74% [35]), belumosudil (67% [10], 78% [43], and 43% [45]), and axatilimab (79% [11]). Lastly, eye response rates are likely to be affected by baseline eye scores, but baseline eye scores were not reported in the studies of ruxolitinib. The response rates in each organ according to agents are shown in Appendix A.

## 4. Summary and Future Directions

Information regarding the expected response rate in individual organs may help the drug choice for patients with refractory chronic GVHD. We summarize these rates for each drug and discuss the caveats in the interpretation of the results. The results should be carefully interpreted because the rates were extracted from studies of chronic GVHD overall but not from studies targeting GVHD in specific organs. Randomized studies designed specifically for individual organs will be needed in order to draw robust conclusions. Furthermore, the limited number of cases for individual organs should be considered.

This review highlights several future directions and unmet needs for the evaluation of organ responses after the treatment of chronic GVHD. First, future studies could be more informative if response rates according to the lines of therapies are reported, and if both the best response rates and fixed-point response rates are reported. The only study with data for both endpoints is the US FDA analysis of its approval of ruxolitinib for the treatment of chronic GVHD after the failure of one or two lines of systemic treatment [51]. This analysis adjudicated the data provided by the sponsor and concluded that the overall response rate at week 24 was 42%, while the best overall response before week 24 was 70%. Among the participants with a response, the median duration of response calculated from the first response to progression, death, or new systemic therapy for chronic GVHD was 4.2 months, and 40% of the responses were lost before 24 weeks. Whether short response durations reflect the limitations of ruxolitinib in the treatment of steroid-refractory chronic GVHD or whether they result from premature tapering of corticosteroid doses is unclear.

Other unresolved questions are whether the duration of the response should be assessed as worsening from the disease severity at the first response or from the baseline and whether reversion to the baseline severity associated with steroid tapering should be counted as the end of a response if the severity is not worse than baseline and if increased steroid doses lower than the baseline reinduce a response. As a less complex alternative, the duration of response can also be measured as the median time to death or new systemic therapy, which was 25 months in the US FDA review of ruxolitinib. The two approaches suggested by the FDA bracket the shortest and longest reasonable ways to define the duration of response.

Second, we observed large discrepancies in the response rates in sclerotic skin according to the current NIH criteria [52] and in the improvement rates according to the patient-reported outcomes for cutaneous sclerosis. Our observations highlight the need to include the proportions of patients with cutaneous sclerosis at baseline in the reports of future studies. The field would benefit greatly from more objective and quantitative measures of cutaneous sclerosis. As one example, expert dermatologists have explored whether a myotonometer could be used to measure the biomechanical parameters of sclerotic skin in patients with chronic GVHD [53].

Third, ocular GVHD is recognized as another highly morbid form of chronic GVHD [48]. Although the best response rates for eyes with ruxolitinib were substantially higher than those with other drugs, baseline eye scores were not reported in any studies of ruxolitinib, which made it difficult to interpret the response rates when comparing different studies. Therefore, ocular response rates should be stratified according to the baseline eye scores in future studies. The International Chronic Ocular GVHD Consensus Group (ICCGVHD) has proposed scoring criteria that incorporate full ophthalmologic evaluations [54]. Compared to the simple NIH criteria, such scoring criteria may have a greater sensitivity for measuring treatment-related changes in the activity of ocular GVHD.

Lastly, high response rates for the lung in three retrospective studies with ruxolitinib and in one of the studies of axatilimab may encourage early pre-emptive intervention in order to prevent the progression of irreversible disease. Moreover, the combined use of ruxolitinib and axatilimab may have synergistic therapeutic potential due to their complementary drug mechanisms and non-overlapping side effects.

## Figures and Tables

**Figure 1 cells-14-00238-f001:**
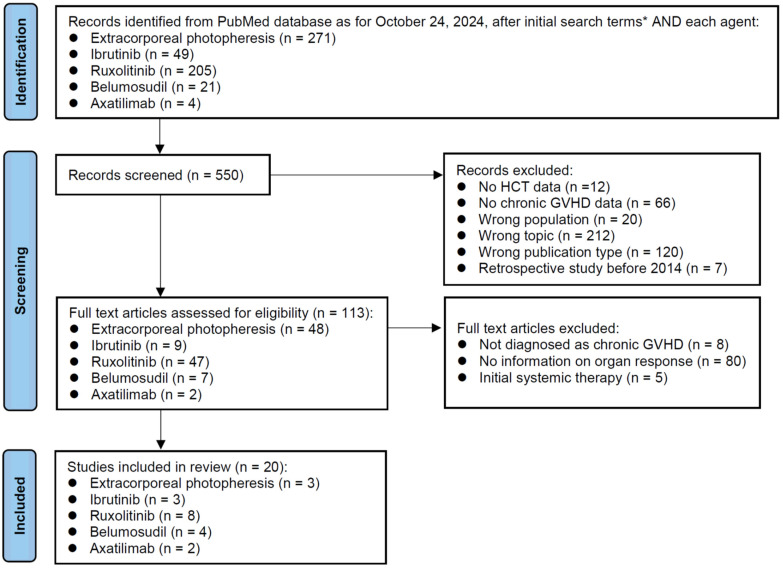
PRISMA diagram. This flow diagram illustrates the review process for study identification, screening, eligibility assessment, and inclusion. Each box provides the number of studies at that stage, and arrows indicate the flow between stages. * Initial search terms: Search: “Graft versus host disease” AND “Date-Publication from 1 January 2005 to 24 October 2024” NOT “review” NOT “case report” NOT “systemic review” NOT “meta-analysis”. Filters: Full text, English.

**Figure 2 cells-14-00238-f002:**
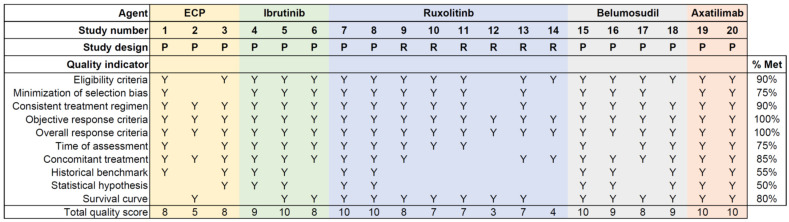
Quality assessment for individual reports. This figure shows the total number of quality criteria met by each of the 20 reports selected from the literature review. Y indicates the criteria met by each report, and %Meet indicates the percent of reports that met each criterion. ECP = extracorporeal photopheresis; P = prospective; R = retrospective.

**Figure 3 cells-14-00238-f003:**
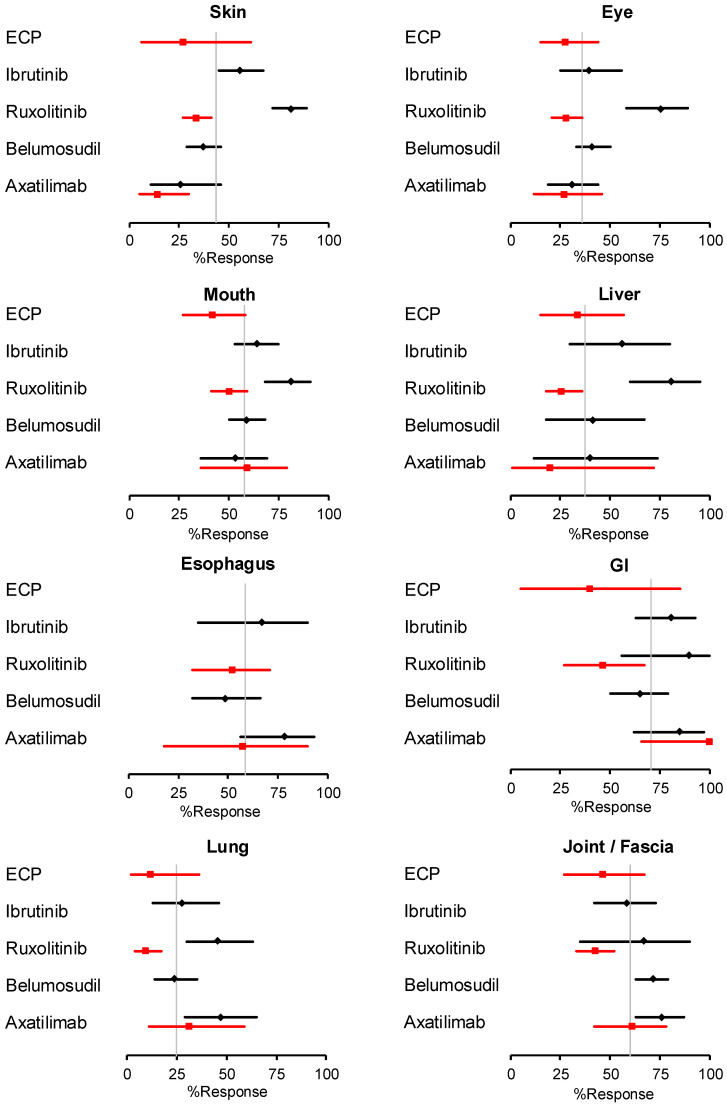
Pooled response rates at 24 weeks for individual drugs according to organs. Best response rates are shown in black and response rates at a fixed point of 24 weeks are shown in red. Diamonds and squares represent mean values with bars showing 95% confidence intervals. The vertical grey line indicates overall means of all drugs regardless of which endpoint was used. ECP = extracorporeal photopheresis, GI = gastrointestinal. These met each criterion.

**Table 1 cells-14-00238-t001:** Summary of selected reports.

#	Agent	% Severe Cases	Median Prior LOTs (Range)	% 2nd Line Treatment	Response Criteria	Response Rate *, % (# of Patients)								
						Overall	Skin	Eye	Mouth	Liver	Esophagus	GI	Lung	Joint/Fascia
1	ECP [6,31]	94	NA	NA	Physician assessment	NA	40 ** (48)	26 ^†^ (27)	43 ^†^ (30)	29 ^†^ (14)	NA	50 ^†^ (2)	22 ^†^ (9)	50 ^†^ (18)
2	ECP [32]	NA	NA	NA	Physician assessment	73 ^‡^ (88)	83 ^‡^ (52)	NA	NA	NA	NA	NA	27 ^‡^ (12)	NA
3	ECP [7]	67	NA	NA	NIH & changes in steroid dose	67 ^†^ (15)	27 ^†^ (11)	31 ^†^ (13)	38 ^†^ (13)	43 ^†^ (7)	NA	33 ^†^ (3)	0 ^†^ (8)	38 ^†^ (8)
4	Ibrutinib [8,33]	NA	2 (1–3)	40	2005 NIH	69 (42)	84 (25)	NA	88 (25)	67 (3)	NA	82 (11)	NA	NA
5	Ibrutinib [34]	47	2 (1–3)	42	2014 NIH	74 (19)	36 (14)	11 (9)	36 (14)	100 (2)	40 (5)	80 (5)	14 (7)	60 (10)
6	Ibrutinib (Pediatric) [35]	74	2 (1–12)	NA	2014 NIH	77 (47)	47 (47)	47 (34)	59 (37)	45 (11)	86 (7)	80 (15)	31 (26)	58 (33)
7	Ruxolitinib [9]	59	1 (1–1)	100	2014 NIH	76 (165)	41 ^†^ (119)	26 ^†^ (96)	50 ^†^ (96)	24 ^†^ (86)	50 ^†^ (18)	40 ^†^ (20)	9 ^†^ (70)	38 ^†^ (45)
8	Ruxolitinib (Sclerotic) [36]	85	3 (2–6)	0	2014 NIH	47 ^†^ (47)	19 ^†^^§^ (47)	38 ^†^ (32)	53 ^†^ (17)	36 ^†^ (14)	56 ^†^ (9)	67 ^†^ (6)	11 ^†^ (9)	45 ^†^ (47)
9	Ruxolitinb [37]	76	3.7 ^¦^ (1–8)	NA	2014 NIH	48 ^†^ (46)	28 ^†^ (39)	22 ^†^ (23)	47 ^†^ (15)	NA	NA	NA	10 ^†^ (10)	46 ^†^ (23)
10	Ruxolitinb [38]	66	3 (1–6)	NA	NIH	71 (41)	79 (28)	57 (14)	76 (29)	80 (15)	NA	100 (2)	50 (18)	56 (9)
11	Ruxolitinb [39]	35	2 (1–6)	NA	2014 NIH	89 (35)	89 (28)	89 (19)	87 (24)	82 (7)	NA	87 (8)	55 (7)	100 (3)
12	Ruxolitinb [40]	38	NA	44	2014 NIH	77 (48)	77 (35)	NA	NA	NA	NA	NA	33 (12)	NA
13	Ruxolitinib + ECP [41]	NA	NA	NA	2014 NIH	74 (23)	44 (18)	20 (10)	NA	21 (14)	NA	54 (13)	13 (8)	100 (1)
14	Ruxolitinib + Belumosudil [42]	86	3 (2–5)	0	2014 NIH	43 (14)	50 (10)	33 (10)	50 (9)	NA	NA	45 (3)	17 (5)	33 (7)
15	Belumosudil [10]	67	3	0	2014 NIH	76 (132)	37 (110)	42 (97)	55 (72)	39 (13)	45 (31)	58 (36)	26 (47)	71 (100)
16	Belumosudil [43]	78	3	NA	2014 NIH	65 (54)	29 (24)	48 (29)	65 (23)	50 (4)	75 (4)	100 (8)	25 (12)	73 (22)
17	Belumosudil [44]	67	3 (1–5)	10	2014 NIH	73 ^¶^ (30)	40 ^¶^ (20)	27 ^¶^ (22)	78 ^¶^ (22)	67 ^¶^ (9)	60 ^¶^ (5)	67 ^¶^ (3)	15 ^¶^ (13)	56 ^¶^ (9)
18	Belumosudil [45]	43	1 (1–4)	57	2014 NIH	86 (21)	56 (11)	20 (15)	67 (18)	NA	50 (2)	50 (2)	0 (6)	80 (5)
19	Axatilimab [22]	85	4 (1–11)	NA	2014 NIH	67 ^†^ (40)	14 ^†^ (35)	27 ^†^ (30)	59 ^†^ (22)	20 ^†^ (5)	57 ^†^ (7)	100 ^†^ (9)	31 ^†^ (16)	61 ^†^ (31)
20	Axatilimab [11]	79	4 (2–12)	0	2014 NIH	74 (80)	27 (64)	31 (59)	53 (40)	40 (10)	78 (23)	85 (20)	47 (32)	76 (55)

* Unless indicated otherwise, the data indicate best response within 24 weeks. ** Response rate at 12 weeks. ^†^ Response rate at 24 weeks. ^‡^ Response rate at completion of therapy. ^§^ 38% improvement by Lee Skin Subscale. 30% improvement by Scleroderma Health Assessment Questionnaire. ^¦^ Mean prior line of therapies. ^¶^ Best response rate with a median follow-up time of 12.9 months. ECP = extracorporeal photopheresis; NA = not available; LOT = line of treatment; GI = gastrointestinal.

**Table 2 cells-14-00238-t002:** Notable adverse events according to each treatment.

Treatment	Adverse Events
Extracorporeal photopheresis	vasovagal reactions, thrombocytopenia, venous access site infection, fatigue
Ibrutinib	muscle cramp, diarrhea, nausea, bleeding/bruising, infection, fatigue, arrhythmia
Ruxolitinb	anemia, thrombocytopenia, neutropenia, diarrhea, infection
Belumosudil	fatigue, headache, infection
Axatilimab	fatigue, infection, lipase elevation, transaminase elevation, creatine phosphokinase elevation, orbital edema

## Data Availability

Not applicable.

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
