# Peer review of "Treatment Response in Individual Organs Affected by Chronic Graft-Versus-Host Disease"

_cells, 2025, doi:10.3390/cells14040238_

Round 1
Reviewer 1 Report
Comments and Suggestions for Authors
It was a pleasure reading this very interesting review; the work is very well outlined and well described.
Here are some observations:
Row 69-71: It could be useful to indicate the rationale for this temporal distinction between prospective and retrospective studies.
Row 133: Table 1: It could be useful to indicate the number of patients, how many steroid-refractory/dependent patients, how many undergoing steroid therapy, the median steroid withdrawal time, the median follow-up of the evaluated studies, the median duration of response, the median achievement of best response, the median duration of response.
For a better reading of the results, it could be useful to associate a graph of the ORR and organ-by-organ responses, divided by drugs.
Row 134: ECP
The schedules are differents: please, indicate the ECP schedules of the studies
Row 151 and the following: in order to uniform the subchapters, it could be useful to indicate the dosage, the ORR, and, subsequently, that organ-by-organ response for each drug
Row 227: Proportions instead of proporions
Therapies are also chosen based on the safety profile:
- Please, indicate in a table a summary of the toxicities of each drugs indicated
Line 217: Pooled response rates are very interesting, however, it is a general response, independent of the number of lines of therapies; it could be useful to carry out the same evaluation but divided into subgroups (first, second or third line)
Conclusions: despite the studies limitations, as indicated, and the need to interpret the data carefully, it would be useful to enrich the conclusions with possible choice suggestions, depending on the setting (organ involved and line of therapy)
Author Response
- Row 69-71: It could be useful to indicate the rationale for this temporal distinction between prospective and retrospective studies.
Response: We thank the reviewer for this suggestion, we have described the rationale statement. (lines 69-70)
- Row 133: Table 1: It could be useful to indicate the number of patients, how many steroid-refractory/dependent patients, how many undergoing steroid therapy, the median steroid withdrawal time, the median follow-up of the evaluated studies, the median duration of response, the median achievement of best response, the median duration of response.
Response: We thank the reviewer for this important suggestion. We understand that the information on the number of steroid refractory/dependent patients, as well as details including dosage, duration of response, median follow-up time, median steroid tapering time, and the median duration of response, is important. However, many reports did not describe these data, making it difficult to include them in the table. Additionally, Reviewer 2 suggested revising Table 1 to make it simpler and easier for readers to understand. We have revised Table 1 accordingly.
- For a better reading of the results, it could be useful to associate a graph of the ORR and organ-by-organ responses, divided by drugs.
Response: According to the reviewer's suggestion, we created a new figure presenting the ORR of each organ, divided by agents. Our purpose for pooled response, however, is for comparing response rates of each agent according to organs. Thus, we included suggested information as Supplemental Figure S1 (lines 262-263).
- Row 134: ECP
The schedules are different: please, indicate the ECP schedules of the studies
Response: In accordance with the reviewer’s suggestion, we have added the ECP schedules in the revised manuscript (lines 136-141).
- Row 151 and the following: in order to uniform the subchapters, it could be useful to indicate the dosage, the ORR, and, subsequently, that organ-by-organ response for each drug
Response: We appreciate the reviewer's suggestion. We have added the dosages and schedules of each agent in the revised manuscript (lines 161-162, 176-177, 202-205, 222-225). Since the overall response rate (ORR) is not the focus of this review, we show it only in Table 1.
- Row 227: Proportions instead of proporions
Response: Thank you for finding a typo. We corrected it on line 254.
- Therapies are also chosen based on the safety profile:
Please, indicate in a table a summary of the toxicities of each drugs indicated
Response: According to the reviewer's suggestion, we have added a new Table 2 summarizing notable adverse events of each drug and explanations (lines 141-142, 162-164, 178-179, 205-207, 225-228).
- Line 217: Pooled response rates are very interesting, however, it is a general response, independent of the number of lines of therapies; it could be useful to carry out the same evaluation but divided into subgroups (first, second or third line).
Response: We agree with the reviewer’s suggestion. Unfortunately, response rates according to lines of therapies were not reported in many reports, and we were not able to perform such subgroup analyses. We have acknowledged this important issue on line 281.
- Conclusions: despite the studies limitations, as indicated, and the need to interpret the data carefully, it would be useful to enrich the conclusions with possible choice suggestions, depending on the setting (organ involved and line of therapy)
Response: As the reviewer kindly pointed out, we tried to enrich the conclusion with possible choice suggestions and mentioned the potential of the preferred choice for organs such as skin, eyes, and lungs. As mentioned in the discussion, however, these results should all be interpreted with caution. Thus, randomized studies designed specifically for individual organs are warranted to draw robust conclusions, as now emphasized in line 276.
Reviewer 2 Report
Comments and Suggestions for Authors
Dear Sir
I have carefully read the paper: Treatment response in individual organs affected by chronic graft-versus-host-disease: A systematic review by Takanobu Morishita and coworkers
In this contribution the authors, on the basis of data obtained from a systematic review of the literature, evaluated the data of patients with chronic GVHD post HCT, become steroid resistant / dependent. The declared aim of the work was to try to establish the efficacy of five second-line therapies (Extracorporeal photopheresis, (ibrutinib, ruxolitinib, belumosudil and axatilimab) on some target organs typically involved in chronic GVHD (skin, eye, mouth, liver, esophagus, gastro intestinal tract, lung, joint and fascia).
I found the article interesting and well written. The selection criteria of the articles to be taken into consideration are well described. I particularly appreciated the flow chart reported in figure 1 and also the quality score of each single article considered in relation to the ten items described in the introduction (figure 2).
I found Table I very complicated and difficult to read and understand and I suggest to drastically revise it for example by eliminating the columns relating to the type of study, the years, severe cases% (which could be indicated in the table captions) and also the quality score which can be omitted since it is reported in figure 2).
The authors report and discuss the observations, highlighting potential limitations of the results.
It might be worth revising the bibliography, inserting, where possible, the indications relating to the DOI.
Author Response
Comment 1: I found Table I very complicated and difficult to read and understand and I suggest to drastically revise it for example by eliminating the columns relating to the type of study, the years, severe cases% (which could be indicated in the table captions) and also the quality score which can be omitted since it is reported in figure 2).
Response 1: We thank the reviewer for this pertinent suggestion. We have revised Table 1 to make it simpler and easier for readers to understand.
Comment 2: It might be worth revising the bibliography, inserting, where possible, the indications relating to the DOI.
Response 2: We have updated the bibliography to include the DOI information.